behaviour, neuroscience

animal emotion, mice, resilience, affective state, depression, cumulative effects

**Author for correspondence:**
Jasmine M. Clarkson
e-mail: jasmine.clarkson@glasgow.ac.uk

†Present address: Institute for Biodiversity, Animal Health and Comparative Medicine, College of Medical, Veterinary and Life Sciences, University of Glasgow, G61 1QH, UK.

# Negative mood affects the expression of negative but not positive emotions in mice

Jasmine M. Clarkson[1,†], Matthew C. Leach[2], Paul A. Flecknell[3] and Candy Rowe[1]

[1]Centre for Behaviour and Evolution, Biosciences Institute, [2]School of Natural and Environmental Sciences, and [3]Comparative Biology Centre, Newcastle University, Newcastle upon Tyne NE1 7RU, UK

JMC, 0000-0001-5899-4245; MCL, 0000-0002-7148-0158; PAF, 0000-0002-1075-1129; CR, 0000-0001-5379-843X

Whether and to what extent animals experience emotions is crucial for understanding their decisions and behaviour, and underpins a range of scientific fields, including animal behaviour, neuroscience, evolutionary biology and animal welfare science. However, research has predominantly focused on alleviating negative emotions in animals, with the expression of positive emotions left largely unexplored. Therefore, little is known about positive emotions in animals and how their expression is mediated. We used tail handling to induce a negative mood in laboratory mice and found that while being more anxious and depressed increased their expression of a discrete negative emotion (disappointment), meaning that they were less resilient to negative events, their capacity to express a discrete positive emotion (elation) was unaffected relative to control mice. Therefore, we show not only that mice have discrete positive emotions, but that they do so regardless of their current mood state. Our findings are the first to suggest that the expression of discrete positive and negative emotions in animals is not equally affected by long-term mood state. Our results also demonstrate that repeated negative events can have a cumulative effect to reduce resilience in laboratory animals, which has significant implications for animal welfare.

## 1. Background

Understanding the emotional capabilities of animals is important across a wide range of disciplines, including animal behaviour, neuroscience, evolutionary biology and animal welfare science [1–3]. Knowing how similar animals' emotional experiences are to our own, and how they underpin behaviour and decision-making, helps to inform the study of emotional disorders in humans and the development of new clinical treatments. In addition, and as importantly, being able to measure negative affective states and moods in animals and know what factors lead to their expression allows interventions to be made to improve animal welfare.

Consequently, most work on animal emotion focuses on negative states, and particularly the alleviation of long-term mood states, such as depression and anxiety [4–6]. These long-lasting moods, or affective states, occur independently of an immediate stimulus and can be differentiated from discrete emotions, which are more acute and stimulus dependent [2]. Although it cannot be known for certain whether or how these negative mood states are consciously experienced in animals, evidence of their existence through a number of different indicators raises significant welfare concerns [7,8].

However, animals can also express positive emotional states, and it is becoming increasingly recognized that these are also relevant to understanding animal welfare [1,9]. Since discrete emotional responses to lifetime events are thought to build cumulatively over time and determine long-term affective states [10–12],

past positive emotions may be just as important as negative emotions in determining an animal's current mood [9,13]. Understanding an animal's capacity to express positive emotions is also important for being fully informed about an animal's welfare state: reduced expression of positive emotions could be seen to be as much of a welfare concern as increased evidence of negative ones [9,14]. Currently, we know very little about the relationship between positive and negative emotions in animals, and about how their expression changes over time with exposure to lifetime events. However, in humans, people who develop major depressive disorder (MDD) in response to stress tend to show blunted positive emotions and stronger negative emotions [15,16]. Consequently, we might expect to see the same in animals. While a few studies have attempted to explore how mood affects the expression of discrete emotions, either the underlying mood state has not been established, or it has been impossible to directly compare the expression of discrete negative and positive emotions because they are measured using different methods [4,17,18].

In this study, we altered the affective states of laboratory mice to better understand how negative mood affects the expression of positive and negative discrete emotions. We used handling method to manipulate their emotional state, since repeatedly handling mice by their tails makes them more anxious and depressed than handling them using a tunnel [19–21]. We employed a single measure, response towards reward, to evaluate both long-term mood and discrete emotions. Human patients with negative mood disorders, including MDD, are less responsive to and experience less pleasure from rewarding stimuli, a symptom commonly referred to as anhedonia [22,23]. Sucrose consumption is a validated measure of anhedonia and depression in mice [24]. However, it is also sensitive to motivational factors (e.g. hunger and satiation [25,26]), so the way in which mice consume sucrose solutions, such as their 'lick cluster sizes', may be a more direct measure of the animals' hedonic 'liking' of a solution [26,27]. Responses towards changes in a sucrose reward can also reveal discrete emotions, both positive and negative [28–32]. Successive contrast paradigms involve exposing animals to unexpected shifts in reward value, typically using a drop (successive negative contrast) or gain (successive positive contrast) in sucrose reward [28–32]. Initially, animals are exposed to a reward of a given value across repeated trials until they learn to associate it with the experimental context. At this point, the reward value is unexpectedly shifted (either increased or decreased), and behaviour in post-shift trials towards the new reward is compared to that of animals that have always received this value of reward [28–32]. Shifted animals exhibit a measurable behavioural response to these changes in reward value which are interpreted as indicating discrete emotional responses akin to 'disappointment' and 'elation', which wane over time [28–34]. Therefore, this approach and methodology provide an opportunity to directly compare the expression of discrete positive and negative emotions in different mood states in an animal model.

## 2. Methods and material

### (a) Animals, housing and husbandry
Sixty-four male C57BL/6 mice (*Mus musculus*) were received from Charles River Laboratories, UK in two batches (32 mice in each, arrival dates 9 May and 4 July 2017) at approximately seven weeks of age. Mice were pair-housed in M2 cages (33 cm (L) × 15 cm (W) × 13 cm (H), North Kent Plastics), with sawdust bedding, nesting material (4HK Aspen chips, NestPak and Sizzlepet nesting, Datesand Ltd, Manchester) and a clear Perspex home cage tunnel (50 mm diameter, 150 mm length). Cages were cleaned once per week. Animals had access to food (Special Diet Services, RM3E diet) and water *ad libitum*, except prior to training and testing for drinking experiments. Animals were maintained on a reverse 12 : 12 hour light/dark cycle (lights off: 10:00–22:00) with all experimental procedures conducted under red light illumination. They were maintained at an optimal temperature of 21 ± 4°C and relative humidity 55 ± 10%. Three days prior to the start of the study, mice were marked for identification on either the shoulder or rump using hair dye (Jerome Russel B Blonde, UK) [19–21].

### (b) Affective state manipulation
Following acclimation, each cage was randomly assigned to one of two treatment groups: tail or tunnel handled. All mice were handled twice daily for 30 s, 60 s apart, for the first 9 days (figure 1). Prior to handling, the nest material and home cage tunnel were removed for 60 s, and care was taken not to disrupt the nest structure. For the tail handling manipulation, the base of their tail was grasped between thumb and forefinger, and the mouse was lifted onto the sleeve of the laboratory coat and held for 30 s. For the tunnel handling manipulation, the mouse was guided into a Perspex tunnel, which was lifted above the cage and held for 30 s [19–21]. Mice were handled by their designated method for routine husbandry and transferring mice for behavioural testing, and prior to the voluntary interaction tests (described below) on days 19 and 27.

### (c) Behavioural measures of affective state
We conducted two validated tests of affective state for laboratory mice at two points in the experiment [35–37] to ensure that our 9-day handling manipulation had been successful in manipulating the animals' affective states prior to and during the contrast experiments (figure 1). We employed two different tests because they rely on the animal's responses towards novelty [35–37]. All behavioural tests were filmed from above (Cube HD 1080, Y-cam) and analysed using Observer XT (v11, Noldus, Virginia, USA).

### (d) Elevated plus maze
On day 10, mice underwent testing in an elevated plus maze (EPM). The EPM arms measured 30 cm (L) × 5 cm (W) with sidewalls of 15 cm on the two closed arms and elevated 50 cm from the ground. Mice were delivered to the centre of the maze facing an open arm and allowed to explore for 5 min. The maze was cleaned between subjects with 70% ethanol. The order in which tail- and tunnel-handled mice were tested was counterbalanced across the day. Due to seven missing datapoints, sample sizes were reduced for statistical analyses (electronic supplementary material, table S1). The number of open arm entries (when all four paws were in the arm), time spent on the open arms and the number of protected stretch attend postures were recorded.

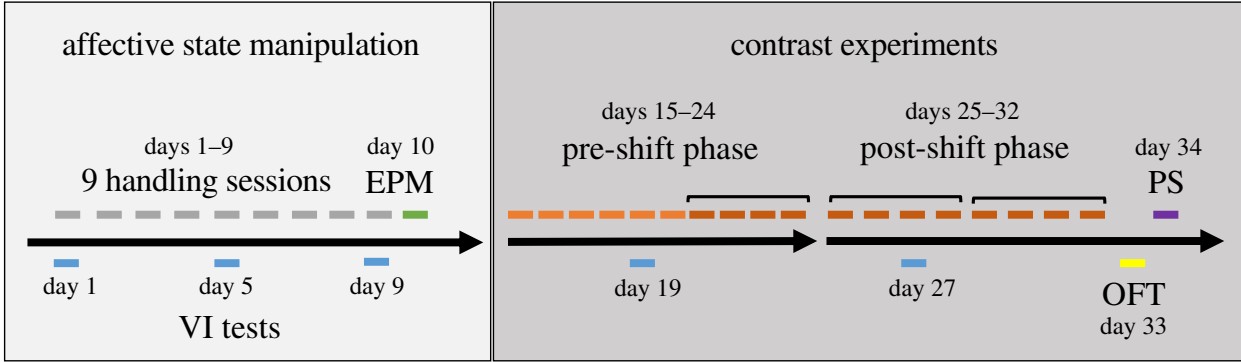

**Figure 1.** The timing and order of the behavioural and physiological measures during the affective state manipulation (grey) and contrast experiments (orange). VI, voluntary interaction tests (blue); EPM, elevated plus maze (green); OFT, open field test (yellow); PS, physiological sampling (purple). The contrast experiments consisted of a 10-day pre-shift phase, followed by an 8-day post-shift phase (dark-orange represent those days used in the main statistical analyses–see full text for details). (Online version in colour.)

**Table 1.** Treatment groups and sample sizes.

| treatment group | pre-shift phase sucrose concentration | post-shift phase sucrose concentration | handling method | sample size |
|---|---|---|---|---|
| successive negative contrast (SNC) | 32% | 4% | tail | 8 |
| | | | tunnel | 8 |
| unshifted (SNC) control | 4% | 4% | tail | 8 |
| | | | tunnel | 8 |
| successive positive contrast (SPC) | 4% | 32% | tail | 8 |
| | | | tunnel | 8 |
| unshifted (SPC) control | 32% | 32% | tail | 8 |
| | | | tunnel | 8 |

## (e) Open field test

On day 33, we conducted an open field test. Each mouse was individually placed in the centre of a rectangular arena (54.5 cm (L) × 35.5 cm (W) × 17 cm (H)) made of white plastic with a transparent Perspex lid and allowed to freely explore for 10 mins. The order in which tail- and tunnel-handled mice were tested was counterbalanced. The total duration spent in the centre, crosses to the centre, total distance travelled, total time spent moving and the velocity of movement for each mouse were recorded.

## (f) Voluntary interaction tests

On designated days during the affective state manipulation (days 1, 5 and 9) and the contrast experiments (days 19 and 27), each cage of animals underwent 'voluntary interaction tests' to assess their responses towards a handler (figure 1) [19–21]. The handler placed either a gloved hand (for tail-handled mice) or a gloved hand holding the home cage tunnel (for tunnel-handled mice) in the home cage for 60 s both pre- and post- handling. Time spent interacting with the handler was recorded for each mouse within a cage, and an overall mean cage score was calculated as a percentage of the total test time (electronic supplementary material table S1).

## (g) Contrast experiments

Mice were trained and tested in eight custom-built drinking chambers. These were standard mice IVC home cages (34

(L) × 19 (W) × 14 (D) cm) with clear Perspex sides, a metal perforated floor and wire cage lid. Solutions were presented through spouts attached to 50 ml falcon tubes on the left-hand side of the cage lid and were weighed before and after each drinking session to determine consumption. Drinking chambers were connected to contact-sensitive dual contact lickometers (Med Associates Inc., St Albans, Vermont), which recorded each lick to the nearest 0.01 s. Custom-written programmes (courtesy of Prof Dominic Dwyer) calculated the lick cluster sizes according to a range of inter-bout intervals (IBIs), which is the length of time between two licks defining when licks can be considered to be in a single bout [26,38]. The data presented in the main text use IBIs of 500 ms (see electronic supplementary material for analysis across other IBIs), meaning that any duration of 500 ms or longer between two licks defined the end of one bout and the start of the next.

Since only eight mice could be tested simultaneously in the drinking chambers, mice in each batch were assigned to four 'testing cohorts'. Testing cohorts were fully balanced with respect to handling method and treatment group (table 1). Water bottles on the home cage were removed 2 h prior to testing to motivate the mice to consume the sucrose solutions during testing [39].

Contrast experiments had two distinct phases (figure 1). The pre-shift phase consisted of 10 consecutive days where mice were transferred to the drinking chamber and had access to sucrose for 15 min. This ensured that the mice had sufficient

experience with the sucrose solution to overcome neophobia and become habituated prior to the post-shift phase [34]. For the first three pre-shift sessions, the spout protruded into the cage to facilitate engagement with the task, after which the spout was positioned in line with the cage to reduce accidental contact. Depending on the treatment group, mice had access to either a low (4% w/w sucrose) or high (32% w/w sucrose) reward (table 1). To confirm that mice had become habituated to the procedure, we analysed sucrose consumption and lick cluster sizes across the last four pre-shift trials.

The post-shift phase lasted for 8 consecutive days since contrast effects are relatively short-lived [29,32,40]. Again, the mice had access to either a 4% or 32% sucrose solution for 15 min where the concentration depended on the treatment group (table 1). In this phase, half the groups were shifted from one concentration to the other. These groups formed either the loss or the successive negative contrast (SNC) condition, where animals were shifted from high to low reward and compared to an unshifted control group that remained on the low reward, or the gain or successive positive contrast (SPC) condition, where animals were shifted from low to high reward, and compared to a matched unshifted control group that remained on the high concentration throughout (table 1). The responses of mice undergoing an SNC or SPC were compared with their respective controls (i.e. mice that were unshifted) to identify contrast effects. Since mice drink small amounts, which can increase variability in measures of lick cluster sizes across trials, we calculated the mean consumption and lick cluster sizes for each individual in the first four and last four post-shift trials [32,34] for use in our statistical analyses. These phases are referred to as post-shift 1 and post-shift 2.

### (h) Physiological measures

Upon completion, animals were humanely killed via cervical dislocation and the adrenal and thymus glands were excised and weighed from the unshifted control animals to investigate whether tail handling has chronic effects on neuroendocrine responses [41].

### (i) Statistical analyses

Analyses were conducted using R studio software [42]. Linear mixed models (see electronic supplementary material, table S1) were fitted using maximum-likelihood estimation. We used a likelihood-ratio test (LRT) between models, which calculates the difference in model deviance ($\chi^2$ distributed) when a predictor variable is removed. Where significant interactions were found, post hoc $t$-tests were performed to determine the factors contributing to the interaction. All statistical tests are reported in electronic supplementary material, table S1.

## 3. Results

The affective state manipulation was successful at inducing differences in anxiety, stress and depression, evident in different measures across standard behavioural tests (figure 2). Tail-handled mice showed less voluntary interaction with the handler than tunnel-handled mice on days 1, 5 and 9, which was maintained throughout the contrast experiments on days 19 and 27 (figure 2a; electronic supplementary material, tables S2 and S3). The differences in voluntary interaction between tail- and tunnel-handled mice were seen on day 1,

which has been previously reported [19–21], and is probably mediated by mice's natural avoidance of predatory threat.

Tail handling also increased measures of anxiety in the EPM. Tail-handled mice showed a lower number of open arm entries, reduced time spent on the open arms and increased protected stretch attend postures (considered to reflect risk assessment behaviours; [43]) compared to tunnel-handled mice (figure 2b). The anxiety-inducing effects of tail handling were long-lasting as seen in the open field test carried out at the end of the contrast experiment (figure 1). Tail-handled mice reduced the time they spent and their frequency of visits to the centre of the open field compared to tunnel-handled mice (figure 2c; also electronic supplementary material, figure S1).

For the first time, we also linked tail handling to a chronic stress response. Tail-handled mice had larger adrenal glands controlled for bodyweight (although no differences in bodyweight were observed; electronic supplementary material, figure S2) than tunnel-handled mice (figure 2d). This suggests that tail handling also has a long-lasting physiological impact [41].

Furthermore, we found evidence that tail-handled mice were in a depressive-like state. Traditionally, anhedonic animals drink less sucrose and have smaller lick cluster sizes than animals considered to be in more positive affective states [5,21,24,26]. When assessing the responses of all mice at the pre-shift phase, we found that tail-handled mice had smaller lick cluster sizes than tunnel-handled mice, but did not find a significant difference in their sucrose consumption (figure 2e; also see electronic supplementary material, S1.2.5–1.2.6). This was surprising since both these measures are considered to reflect the animals' hedonic responses towards sucrose [5,21,26,27,38]. Therefore, we conducted a further analysis to look for differences in sucrose consumption between our tail- and tunnel-handled mice across the two subsequent post-shift phases. This could only be conducted for control mice which received the same concentration of sucrose throughout. We found that tail-handled mice did drink less sucrose than tunnel-handled mice across the two post-shift phases (figure 2f; see also electronic supplementary material, table S5). Therefore, tail- and tunnel-handled mice did show differences in both measures of anhedonia during the experiment.

### (a) Contrast experiments

In order to make inferences with regard to the effects of contrast, it was important that sucrose consumption and lick cluster sizes reflected the predicted difference in reward value (i.e. the higher reward was valued more). When assessing the responses of all mice at the pre-shift phase, we found that while lick cluster sizes were indeed higher for the more rewarding 32% sucrose solution, consumption did not accurately reflect the difference in reward value (figure 2e,f). Mice drank more of the 4% than the 32% sucrose solution (see electronic supplementary material, section S1.2.5), most likely because of the higher satiating properties of 32% sucrose solution [44]. Therefore, we focused on changes in lick cluster size to measure the responses of mice towards changes in reward value in the contrast experiments.

Interestingly, post hoc analyses comparing each individual contrast group with their respective control group at the pre-shift phase did not always reveal a significant concentration effect (see electronic supplementary material, S1.3.1). However, given that we observed measurable changes in

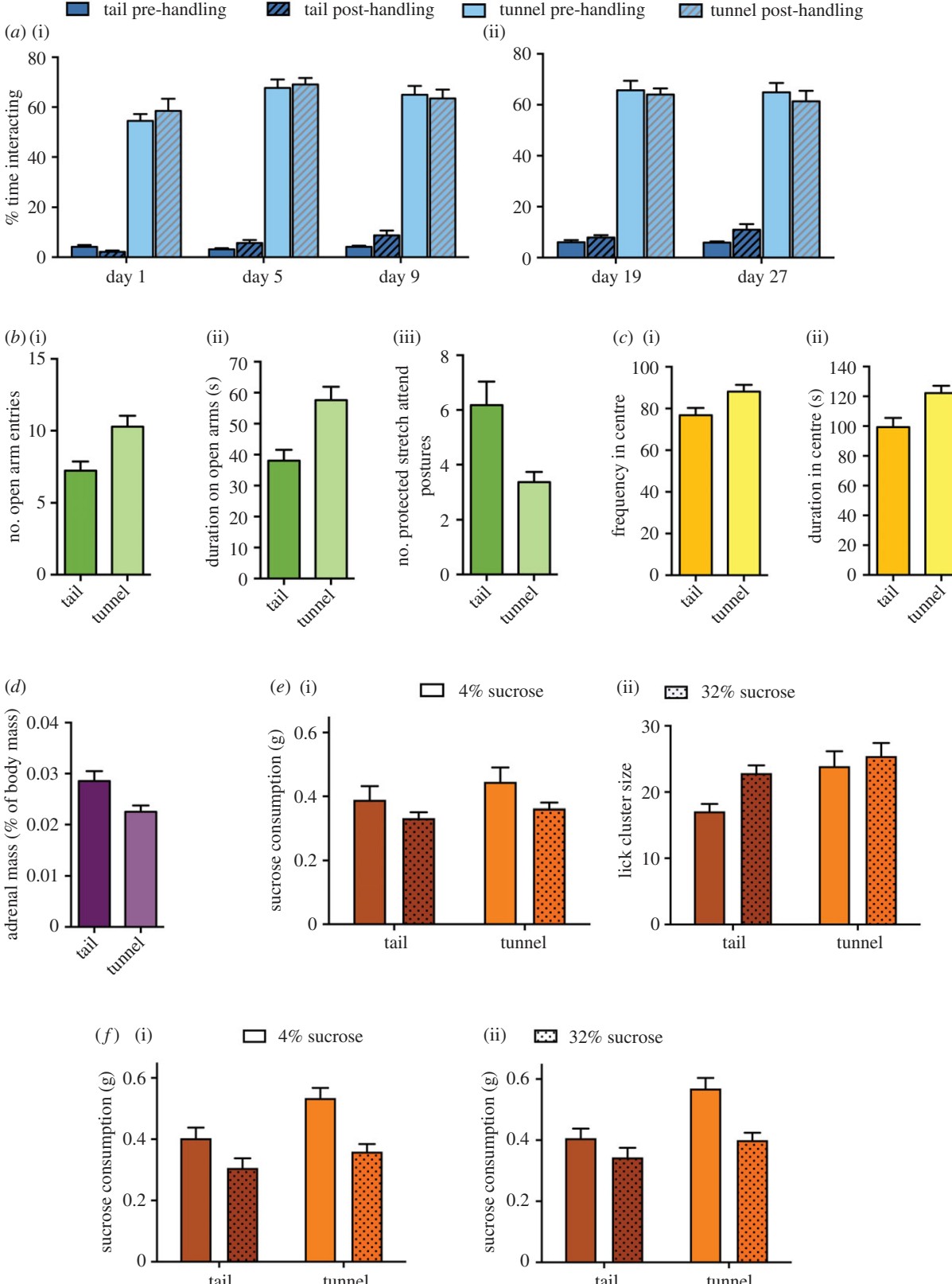

**Figure 2.** Significant differences in behavioural and physiological measures showed that the tail-handled mice were more anxious, depressed and chronically stressed than the tunnel-handled mice. (*a*) In the voluntary interaction tests which assessed mices' responses towards a gloved hand or gloved hand holding a tunnel, (i) tail-handled mice spent less time interacting with the handler during the affective state manipulation period ($\chi^2 = 62.13$, $p < 0.001$), an effect which (ii) remained evident during the contrast experiments ($\chi^2 = 52.20$, $p < 0.001$; see electronic supplementary material, tables S2 and S3). (*b*) Tail-handled mice (i) visited the open arms of the elevated plus maze less often ($t_{54.1} = 3.21$, $p = 0.002$) and (ii) spent less time there ($t_{53.6} = 3.60$, $p < 0.001$), and (iii) showed more protracted stretch attend (PSA) postures ($t_{44.4} = 2.06$, $p = 0.046$). (*c*) Tail-handled mice (i) visited the centre of the open field test less often ($t_{61.6} = 2.36$, $p = 0.021$) and (ii) spent less time there ($t_{58.7} = 2.94$, $p = 0.005$). (*d*) The adrenal glands of tail-handled mice were larger than those of tunnel-handled mice ($t_{28} = 2.36$, $p = 0.025$). (*e*) (i) Tail-handled mice did not consume less sucrose than tunnel-handled mice at the end of the pre-shift phase ($F_{1,60} = 1.63$, $p = 0.207$). (ii) Tail-handled mice did have smaller lick cluster sizes than tunnel-handled mice ($F_{1,60} = 4.39$, $p = 0.040$; see electronic supplementary material sections 1.2.5–1.2.6). (*f*) Tail-handled control mice consumed less sucrose than tunnel-handled control mice during the contrast experiments in both (i) post-shift 1 and (ii) post-shift 2 phases ($\chi^2 = 5.44$, $p = 0.019$; see electronic supplementary material, section S1.2.7). (Online version in colour.)

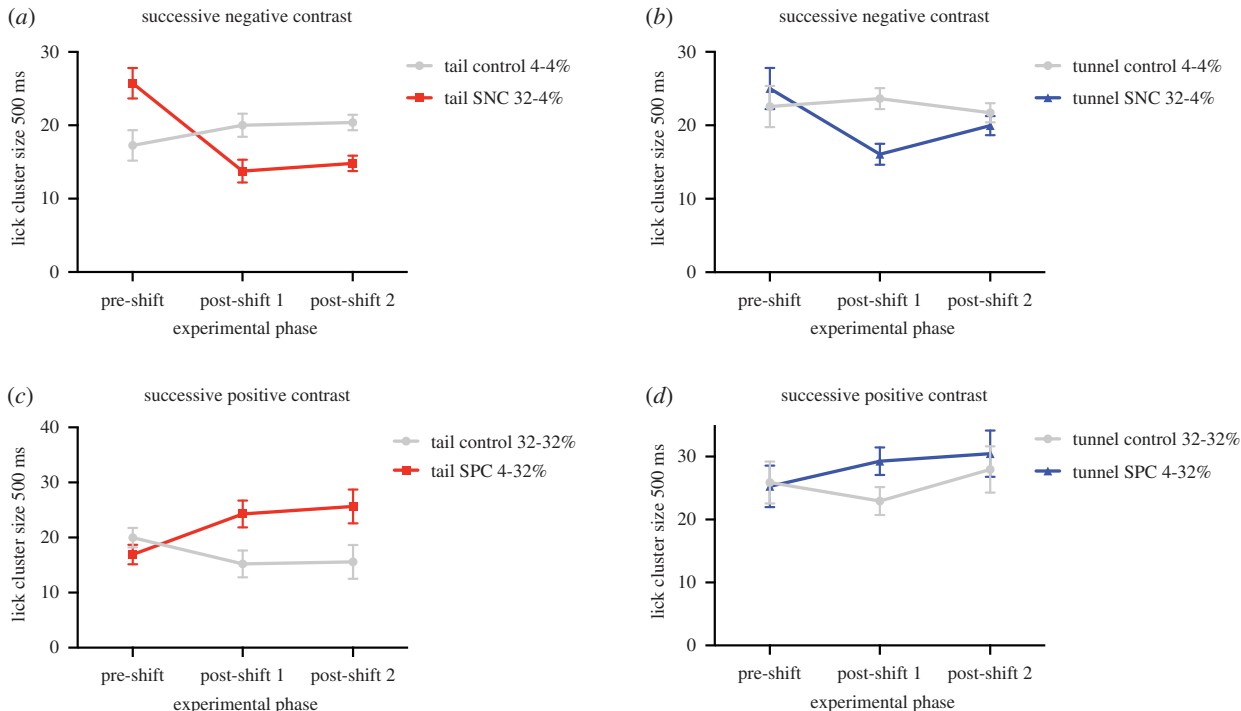

**Figure 3.** Mean (±SEM) lick cluster size using an inter-bout interval of 500 ms for tail and tunnel-handled mice undergoing the four contrast conditions. (*a*) Tail-handled mice and (*b*) tunnel-handled mice undergoing the SNC contrast condition (32-to-4% sucrose) compared to control mice that received 4% sucrose throughout the pre-shift and post-shift phase. (*c*) Tail-handled mice and (*d*) tunnel-handled mice undergoing the SPC contrast condition (4-to-32% sucrose) compared to control mice that received 32% sucrose throughout the pre-shift and post-shift phases. (Online version in colour.)

lick cluster size following changes in reward value at the post-shift phase, and a clear concentration effect at the pre-shift phase when assessing the responses across all mice, we were confident that our mice were capable of detecting the difference between our two reward values.

We also established that control mice had stable lick cluster sizes prior to the post-shift phases in order to provide meaningful comparisons with animals undergoing contrast conditions. Importantly, when comparing lick cluster sizes across the last four trials of the pre-shift phase, we found no significant effect of the trial on lick cluster sizes of control mice (all $F \leq 2.96$, all $p \geq 0.095$; see electronic supplementary material, section S1.3.2), suggesting these mice had stabilized their behaviour towards sucrose.

### (b) Successive negative contrast

For mice undergoing the SNC contrast where the reward value decreased from 32% sucrose to 4% between the pre-shift and post-shift phases, we found that tail-handled mice continued to have smaller lick cluster sizes in the two post-shift periods compared to tunnel-handled mice ($\chi^2 = 7.56$, $p = 0.006$). Furthermore, both tail- and tunnel-handled mice that were shifted from a high to a low reward demonstrated a negative contrast effect ($\chi^2 = 18.37$, $p < 0.001$). However, this contrast effect differed across the two post-shift periods depending upon the method by which mice were handled (significant three-way interaction between handling method, post-shift period and contrast condition $\chi^2 = 4.10$, $p = 0.043$; see electronic supplementary material, table S6). Independent *t*-tests revealed that in the first post-shift period, both tail- and tunnel-handled mice shifted from a high to a low reward had smaller lick cluster sizes compared to mice that had remained on the lower 4% sucrose solution throughout (tail handled:

$t_{(61.15)} = 4.33$, $p < 0.001$; figure 3*a*; tunnel handled: $t_{(50.87)} = 5.33$, $p < 0.001$; figure 3*b*). However, this was not the case at the second post-shift period, where compared to their respective controls, only tail-handled mice continued to have smaller lick cluster sizes at the second post-shift period ($t_{(61.87)} = 4.46$, $p < 0.001$; figure 3*a*), whereas tunnel-handled mice did not ($t_{(59.97)} = 1.14$, $p = 0.261$; figure 3*b*). These findings were not exacerbated by drift among the control animals, since there was no significant difference in lick cluster sizes across the two post-shift phases for tail and tunnel control mice (tail control: $t_{31} = 0.33$, $p = 0.737$; tunnel control: $t_{31} = 1.07$, $p = 0.292$). Therefore, taken together, tail-handled mice showed a longer-lasting negative contrast effect than mice handled using tunnels.

### (c) Successive positive contrast

For mice undergoing positive contrast, receiving a low (4%) concentration of sucrose in the pre-shift phase followed by a high (32%) sucrose solution in the post-shift phase, we found that both tail- and tunnel-handled mice showed a strong contrast effect ($\chi^2 = 7.17$, $p = 0.007$; figure 3*c*,*d*). Therefore, irrespective of their handling experience, mice upshifted from 4% to 32% sucrose had significantly larger lick cluster sizes than animals receiving the 32% sucrose throughout both phases (figure 3*c*, *d*). However, although we found that tunnel-handled mice had larger lick cluster sizes than tail-handled mice overall ($\chi^2 = 8.14$, $p = 0.004$), in contrast with our data for SNC, we found no evidence that this differed across the post-shift periods between our contrast conditions depending on the method by which mice were handled (non-significant three-way interaction $\chi^2 = 1.03$, $p = 0.311$). There were also no other significant main effects or interactions between these factors (all $p > 0.05$; see electronic supplementary material, table S7).

## 4. Discussion

We successfully induced negative affective states within tail-handled animals, which were more anxious and depressed than mice handled using tunnels [19–21]. Tail-handled mice spent less time in the open arms of the EPM and centre of the open field, and we found evidence for reduced sucrose consumption and smaller lick cluster sizes across our contrast experiment. We also found that tail-handled mice were more sensitive to reward loss (i.e. they showed more sustained 'disappointment') although their sensitivity towards reward gain (i.e. their capacity to show 'elation' [28–32]) remained unaffected. This suggests that negative mood states affect the expression of discrete emotions towards external events, and specifically, the capacity to be resilient towards negative ones. Our findings have implications for current thinking in understanding animal emotions and welfare.

Our finding that tail handling, a laboratory procedure shown to induce negative affect [19–21], made mice more prone to disappointment is in itself not surprising. In the clinical literature, depressed patients often report changes in the way that they pay attention to, perceive and make judgements and decisions about the world around them [15,45]. Typically, people in negative affective states pay more attention to negative events and perceive them as being worse than individuals in a more positive affective state [15,45,46]. Furthermore, people with MDD report greater disappointment towards reward losses than non-depressed healthy controls [47,48]. The one study investigating disappointment in animals in a welfare context also found that chronic exposure to negative conditions can enhance sensitivity to reward loss: laboratory rats have a prolonged SNC when living in unenriched compared to enriched caging [32]. While this study used a manipulation considered to change animals' affective state, the authors did not confirm this through any associated behavioural or physiological measures. Our study can more conclusively link negative affective state (anxiety and depression) with an enhanced sensitivity to reward loss and demonstrates the similarities between animal and human emotions [3].

By contrast, being in a negative mood state did not appear to influence animals' sensitivities to reward gain, with both tail- and tunnel-handled animals showing sustained positive contrast effects. This is an intriguing and unexpected result, since the tail-handled mice showed blunted responses towards the sucrose solution and were more anhedonic overall, which is a core symptom of clinical depression. Although this may seem counterintuitive, patients in some studies of MDD show similar increases in positive affect towards positive events when compared with less depressed individuals [49,50]. In addition, like our mice, these patients also show sustained disappointment towards negative events [50]. Therefore, our data support the idea that the capacity to express positive emotions might be unaffected by negative mood states, unlike the expression of negative emotions. This affects our understanding of animals' abilities to experience positive as well as negative emotions, and how they might build cumulatively over time to affect mood and impact on an animal's welfare [10,11].

Our approach of using responses towards reward allowed us to measure both long-term changes in mood and directly compare the expression of discrete positive and negative emotions in animals for the first time. We found that consumption did not reliably reflect hedonic valuation: the higher concentration was consumed less than the lower concentration (probably due to the satiating effects of sucrose; [44]), and we only detected reduced consumption in our tail-handled mice in the post-shift phases. In comparison, lick cluster size was a more robust and sensitive measure of hedonic valuation of reward, capable of detecting both long-term and short-term emotional changes. Overall, mice preferred 32% over the 4% sucrose, and tail-handled mice were more anhedonic than tunnel-handled mice even by the end of the pre-shift phase. Although our retrospective analyses of each individual pair of treatment groups in the pre-shift phase did not consistently reveal a significant preference for 32% over 4% sucrose solution, we are confident that both our pooled data and the clear contrast effects across all treatment groups in the post-shift show that mice can detect and evaluate differences in our rewards. Differences in our contrast effects were not attributable to changes in behaviour in our control groups, which showed stable lick cluster sizes at the pre-shift phase, and no evidence of drift during the post-shift phases. Therefore, although sucrose consumption is often used as a single measure of anhedonia, lick cluster size could be a valuable measure for understanding more about the emotions of laboratory mice and be a useful tool in evaluating their welfare [21].

Animal welfare studies exploring the emotional states of animals typically focus on the measurement and avoidance of negative affective states, such as fear, anxiety and depression [1,9]. However, there is a growing appreciation that good welfare should also make sure that animals express discrete positive emotions, such as 'happiness' or 'joy' [9]. There has been very little work in this area [51–53], and indeed, to our knowledge, this is the first evidence of positive emotions akin to 'elation' in mice. Nonetheless, what our data indicate is that animals are capable of expressing positive emotions despite being in negative long-term mood states, and that measuring positive and negative emotions is important. If we had measured how mice responded to a positive event only, it would have been tempting to conclude that the effects of tail handling are not a welfare concern, which we know is not the case. Of course, we might have seen differences between our treatment groups, if we had continued our experiment for longer, or perhaps other measures of positive affect are more influenced by negative mood state [51,54]. How and when animals experience positive affect, and what factors influence it, is a fruitful area for future research, and our work highlights the importance of simultaneously measuring both positive and negative affect.

Our findings extend the growing literature on the negative effects of tail handling on mouse welfare [19–21,55] by showing that as well as being more anxious and depressed [19–21], tail-handled mice were also less resilient to negative events. Laboratory conditions contain many potential stressors, with animals held under controlled conditions and often subject to a range of scientific procedures associated with pain and distress to varying degrees. Therefore, the suggestion that tail-handled mice may perceive such potentially stressful and aversive events more negatively and, for longer, is particularly concerning. If animal welfare is influenced by the cumulative effects of positive and negative events that an animal experiences in its lifetime [10,11], then tail handling could make negative events and experiences build cumulatively across time quicker and to a greater extent compared to tunnel handling. If this is the case, tunnel handling of

laboratory mice should become more widely adopted to make mice more resilient to negative events and improve their welfare.

We also show conclusively that tail handling causes long-term physiological effects associated with chronic stress [56]. Tail-handled mice had larger adrenal glands compared to tunnel-handled mice, suggesting hyperactivity of the hypothalamic–pituitary–adrenal (HPA) axis in response to long-term exposure to stress [41,57]. This is not only concerning from an animal welfare point of view, but also a scientific perspective. It is important to understand the full impact of tail handling on physiological processes to understand how the results from biomedical research could be affected by handling method, and if tunnel handling could lead to more clinically robust and reproducible data [21,55,58].

Our findings offer unique insights into the emotional lives of laboratory mice and show that mice in a negative affective state are capable of experiencing discrete emotions such as disappointment and elation [28–32]. These findings have important implications for understanding how their experiences within the laboratory influence their overall affective state and highlight how little we know about how mood states influence everyday experiences. We encourage further work simultaneously exploring both positive and negative discrete emotions in order to better understand animal emotion and welfare.

Ethics. Experiments were approved by Newcastle University's Animal Welfare and Ethical Review Body (AWERB Project ID: 540) for subthreshold unregulated work in accordance with the EU Directive (2010/63/EU) and ASPA (1986). Animals were checked daily, and no adverse effects were reported.

Data accessibility. All data generated or analysed during this study are available from the Dryad Digital Repository: https://doi.org/10.5061/dryad.x69p8czg4 [60].

Authors' contributions. C.R., M.C.L. and P.A.F. acquired funding for the study, J.M.C. and C.R. conceived the idea, designed the study and protocols and drafted the manuscript, J.M.C. carried out experiments and analysed the data, and all authors commented on the manuscript.

Competing interests. The authors declare no competing financial interests.

Funding. This work was supported by a BBSRC DTP (grant no. BB/J014516/1).

Acknowledgements. We would like to thank the staff at the Comparative Biology Centre at Newcastle University for excellent technical assistance, Dominic Dwyer for providing software and training for J.M.C, and Melissa Bateson and two anonymous reviewers for their constructive comments on the manuscript.

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
