## [Reviewer comments · Proceedings of the Royal Society B: Biological Sciences]

Review History

RSPB-2020-0030.R0 (Original submission)

Review form: Reviewer 1

Recommendation

Accept with minor revision (please list in comments)

Scientific importance: Is the manuscript an original and important contribution to its field?

Excellent

General interest: Is the paper of sufficient general interest?

Excellent

Quality of the paper: Is the overall quality of the paper suitable?

Good

Is the length of the paper justified?

Yes

Should the paper be seen by a specialist statistical reviewer?

Yes

Do you have any concerns about statistical analyses in this paper? If so, please specify them explicitly in your report.

No

It is a condition of publication that authors make their supporting data, code and materials available - either as supplementary material or hosted in an external repository. Please rate, if applicable, the supporting data on the following criteria.

Is it accessible?

N/A

Is it clear?

N/A

Is it adequate?

N/A

Do you have any ethical concerns with this paper?

No

Comments to the Author

This paper addresses a very interesting question - the relationship between longer-term mood and expression of discrete negative/ positive emotions - the paper could be of considerable general interest - the paper's approach is novel and potentially exciting

However I am concerned that the authors are somewhat overstating their findings at least with respect to the claim that they have unequivocally demonstrated that their handling treatments have given rise to expected changes in anhedonic responses - here their results vary in one important respect from their (& others) previous work in that the mice did not reduce their consumption of sucrose as expected.

In response to this the authors turn to another measure (lick cluster size) as their measure of anhedonia which allows them to proceed to claim that the tail-handling treatment induced anhedonia - however as the authors state themselves (lines 271-273) previous work on anhedonia using the sucrose model finds both a reduction in level of sucrose drunk as well as changes in lick cluster size

Given this dissociation of these 2 measures of anhedonia I think we have to view the remaining results on contrast effects with some caution as we cannot be certain of the baseline condition (i.e. whether the tail-handled mice were indeed anhedonic) - the authors do acknowledge some of these issues lines 306-314; here in effect they 'bin' sucrose consumption as a measure of contrast effects because it does not fit their expectations ('consumption did not accurately reflect the difference in reward value'); in doing so they are choosing to not fully consider both their own and others previous findings where tail-handling led to a reduction in sucrose consumption.

One further point on this - eye-balling the response curves to both the negative and positive contrasts the shape of the response is similar for both contrasts and both handling treatments - in both cases visually the tail-handled mice do not return to control levels whereas in both cases the tunnel handled mice do - I know that this is not reflected in the statistics (in terms of the 3 way interaction handling * contrast * phase) but it does perhaps suggest that there is more similarity in response to the contrast conditions than the authors suggest.

One option for revision could be for the authors to be more cautious in their claims and to more fully discuss the points I raise above - another would be to repeat the experiment but this may not be possible for all the obvious reasons -

Another issue is the question of whether the authors are distinguishing between mood and emotional states as they claim

In brief they use the handling treatments to establish a longer-term mood state which they validate with various tests/ measures

They then apply the contrast tests as measures of acute emotional responses ('disappointment'/ 'elation') - however they measure the contrast effect over a number of days (post-shift) and indeed with the negative contrast show longer-term effects of handling on disappointment (with a difference emerging between tail-handled vs tunnel handled emerging some days after the shift) - so my question is given that the contrast effect (as affected by handling) only emerges after some days post-shift can this be regarded as an acute emotional response?

Other points/ suggestions

Line 24: 'Knowing how animals experience emotions is challenging' - I would suggest that it is unlikely we will ever know how animals experience emotions

Lines 43-50: I would have thought that the prime reasons for investigating animal emotions was for the animals themselves and not for comparison to ourselves -

Line 53-54: change positive states to psychological states

Line 55: the quoted references do not provide direct support for this statement

Lines 51-65: I suggest referring to Fraser, D., & Duncan, I. J. (1998). 'Pleasures', 'pains' and animal welfare: toward a natural history of affect. *Animal welfare*, 7(4), 383-396.

Line 80: Refer to Burman's work on contrast effects?

Lines 136-137: As written the handling does not sound entirely standardised with mice potentially having different handling experiences depending on the requirements of their husbandry ('From that point, mice were only handled by their designated method, including for routine husbandry and transferring mice for behavioural testing')

Line 206-208: Explain the purpose of the setting up of testing cohorts

Line 266-269: As indicated above I see this as an 'overstatement' or alternatively not dealing sufficiently with the sucrose consumption issue

Fig 1D: The legend for sucrose consumption should be changed to be more understandable Lick cluster as a measure looks more sensitive at lower sucrose concentration

Lines 306-314: I found that this paragraph was too dismissive of the lack of effect of handling on sucrose consumption

Review form: Reviewer 2

Recommendation

Major revision is needed (please make suggestions in comments)

Scientific importance: Is the manuscript an original and important contribution to its field?

Good

General interest: Is the paper of sufficient general interest?

Good

Quality of the paper: Is the overall quality of the paper suitable?

Good

Is the length of the paper justified?

Yes

Should the paper be seen by a specialist statistical reviewer?

No

Do you have any concerns about statistical analyses in this paper? If so, please specify them explicitly in your report.

No

It is a condition of publication that authors make their supporting data, code and materials available - either as supplementary material or hosted in an external repository. Please rate, if applicable, the supporting data on the following criteria.

Is it accessible?

Yes

Is it clear?

Yes

Is it adequate?

Yes

Do you have any ethical concerns with this paper?

No

Comments to the Author

This paper describes a well-designed study of laboratory mice to investigate the relationship between an induced long-term negative affective state or 'mood' and shorter-term positive and negative affective states ('emotions') inferred from responses made in successive negative contrast (SNC) and successive positive contrast (SPC) tests. The interface between 'moods' and 'emotions' is a theoretically important area of animal emotion research and may have practical implications, including in an animal welfare context as the authors argue. The data are collected and analysed appropriately, the paper is clearly written and the results are interesting - the data provide convincing evidence of induction of the negative mood state, plus the suggestion that such a state may have a moderating effect on SNC and SPC responses. My main comments are on interpretation of results of the SNC and SPC tests and the potential for lack of differences during pre-shift baseline training and drift in the control groups during post-shift testing to have contributed to the findings (see below). I also have comments on clarifying concepts and terminology (always a potential source of confusion in studies of animal emotion), and some methodological details.

Main comments

Lines 213-216: A perennial challenge of SNC and SPC tests is to ensure that, at the time of the reward shift, groups trained on one reward are behaving in a clearly different way to those trained on the other (and hence that the rewards are valued differently), and also that control groups are showing a consistent response across sessions so that post-shift changes can be attributed to changes in treatment group behaviour rather than continuing drift in control groups. Did you check whether control group behaviour 'plateaued' during the pre-shift phase and was different to that of treatment groups? If not, what was the rationale for selecting a 10-day

pre-shift period?

Lines 317-322: The analysis of SNC and SPC data don't appear to include a check on whether the groups trained on different sucrose concentrations within each comparison pair (Fig. 2A-D) show different lick-cluster sizes at the pre-shift (baseline) stage (see above). Although the results of the presented analysis of the pre-shift 'anhedonia test' (Fig. 1D) demonstrate a weak concentration effect, this looks to be primarily driven by the lower lick-cluster size in tail-handled mice exposed to 4% sucrose (though there is no significant interaction effect). Because Figs 2B-D all indicate very close pre-shift baselines, this point and its potential effects on interpretation must be considered in the Discussion.

Line 322-335: Post-hoc analyses of 3-way interactions can be done in several ways and one alternative here, related to the point made above about the possibility of 'control group response drift' following reward shift, would be to investigate whether control groups demonstrated different cross-time shifts in the different groups. For example, it looks from the graphs as if there was some upwards drift in lick cluster size in the tail-handled control group that was not so evident in the tunnel-handled control group, and this could contribute to the observed results. It would strengthen findings to rule this out, and the possibility that it influences findings should be considered in the Discussion.

Lines 344-349: The graphs indicate that tail-handled mice showed a more prolonged SPC than tunnel-handled mice (i.e. at post-shift phase 2; this might again be partly due to drift in control group behaviour, e.g. downwards in tail-handled controls), so it is perhaps surprising to find no statistical effect. Given that there is also no 3-way interaction, it is difficult to justify further investigation of the data but some consideration of this possibility, including the influence of control group drift, would be useful to see in the Discussion.

Additional comments

Lines 33,35,401,403,413,457 and throughout: The word 'experience' as in 'experience discrete positive emotions' needs qualification because it suggests a conscious experience of the state and it is difficult / impossible to measure such conscious experiences in non-humans. Clarification of exactly how this term is being used in the paper will help avoid readers being confused about what is being proposed or inferred by the authors. The same comment applies for terms such as 'emotion', 'mood', 'depression', 'anxiety', 'disappointment' and 'elation' - a general statement about terminology is probably the most efficient way of clarifying usage (e.g. that these states are inferred from particular indicators, but that currently we cannot know for certain whether and how they are consciously experienced).

Lines 61-69: Please provide a brief explanation of why (theoretically and/or empirically) it is interesting to know about the interplay between 'mood' and 'emotion'. Specific hypotheses would also be useful if appropriate.

Lines 77-79: I think it would be more accurate to say here that the animal's discrete emotional responses can be inferred from measured responses to changes in reward value.

Lines 86-89: The 'discrete emotional response' doesn't indicate 'disappointment' or 'elation', it is the behavioural response that is used to infer these states which are themselves the discrete emotional responses. Again, this could be rephrased more clearly to say that particular behavioural responses to a gain or drop in reward value (the responses should be briefly described) can be interpreted as indicating a 'disappointment-like' or 'elation-like' state.

Line 90: Add a reference to a mouse SNC or SPC task.

Line 133: Please refer to the timeline of the study shown in Fig. 1A before starting to describe the tests etc. I was hoping for such a figure throughout the methods description as I tried to work out

how everything fitted together!

Line 183: I assume that the 'sucrose drinking tests' refers to the 'contrast experiments'? If so, please keep terminology consistent.

Line 192: What days were the contrast experiments carried out on?

Line 221: The text says that there were 8 post-shift days, but Fig. 1A indicates only 4? In fact, it's not clear exactly when the 18 days of the contrast experiments occur on Fig. 1A. Please clarify.

Lines 222-223: Briefly explain why you divided these data into two bins, and what was the outcome measure in each bin (e.g. the mean of the responses shown in each 4-day block?).

Table 2: Please clarify what 'Anhedonia tests' refers to as there is no specific mention of this in the text or Fig. 1A. Is this the first (pre-shift) 10 days of the contrast experiments? If so, terminology should be used consistently to avoid confusion. Also, for the Open Field and Adrenal Weight measures, mice will have experienced not just the handling treatment but also the contrast treatment but this is not included as a factor in the statistical models – please provide justification of this.

Fig. 1B: It is interesting that the pre-handling time spent interacting with the hand is very low for tail-handled mice on day 1 even before the treatment has started. If this is correct, it needs to be considered in the Discussion (e.g. was there some (unintended/random) difference between groups prior to the start of the study). If there is a good reason for this (e.g. handling was actually imposed before the start of the Affective State Manipulation period; mice are 'naturally' avoidant of human hands), this needs to be made clear.

Lines 276-281: Please add a reference to the Supplementary Materials to indicate that other results from the open field and physiological measures can be found there.

Lines 279-280: Clarify that the adrenal weight measure is a percentage of body weight, and that no significant difference in body weight was detected (Supplementary Materials).

Line 386: Might one expect different effects of anxiety and depression-like mood states on short-term emotions and, if so, could this also explain some results?

Decision letter (RSPB-2020-0030.R0)

29-Feb-2020

Dear Dr Clarkson:

I am writing to inform you that your manuscript RSPB-2020-0030 entitled "Negative mood affects the expression of negative but not positive emotions in mice" has, in its current form, been rejected for publication in Proceedings B.

This action has been taken on the advice of referees, who have recommended that substantial revisions are necessary. With this in mind we would be happy to consider a resubmission, provided the comments of the referees are fully addressed. However please note that this is not a provisional acceptance.

The resubmission will be treated as a new manuscript. However, we will approach the same reviewers if they are available and it is deemed appropriate to do so by the Editor. Please note

that resubmissions must be submitted within six months of the date of this email. In exceptional circumstances, extensions may be possible if agreed with the Editorial Office. Manuscripts submitted after this date will be automatically rejected.

Sincerely,
Dr Sasha Dall
mailto: proceedingsb@royalsociety.org

Associate Editor
Board Member: 1
Comments to Author:

Both reviewers found the manuscript well presented and of good general interest. They have identified several key areas that require attention, in particular: further clarification of concepts and terminology; more cautious and more detailed consideration of the results in relation to previous studies; and potential additional statistical analysis to consolidate interpretation of the study findings.

Reviewer(s)' Comments to Author:

Referee: 1

Comments to the Author(s)

This paper addresses a very interesting question - the relationship between longer-term mood and expression of discrete negative/ positive emotions - the paper could be of considerable general interest - the paper's approach is novel and potentially exciting

However I am concerned that the authors are somewhat overstating their findings at least with respect to the claim that they have unequivocally demonstrated that their handling treatments have given rise to expected changes in anhedonic responses - here their results vary in one important respect from their (& others) previous work in that the mice did not reduce their consumption of sucrose as expected.

In response to this the authors turn to another measure (lick cluster size) as their measure of anhedonia which allows them to proceed to claim that the tail-handling treatment induced anhedonia - however as the authors state themselves (lines 271-273) previous work on anhedonia using the sucrose model finds both a reduction in level of sucrose drunk as well as changes in lick cluster size

Given this dissociation of these 2 measures of anhedonia I think we have to view the remaining

results on contrast effects with some caution as we cannot be certain of the baseline condition (i.e. whether the tail-handled mice were indeed anhedonic) - the authors do acknowledge some of these issues lines 306-314; here in effect they 'bin' sucrose consumption as a measure of contrast effects because it does not fit their expectations ('consumption did not accurately reflect the difference in reward value'); in doing so they are choosing to not fully consider both their own and others previous findings where tail-handling led to a reduction in sucrose consumption.

One further point on this - eye-balling the response curves to both the negative and positive contrasts the shape of the response is similar for both contrasts and both handling treatments - in both cases visually the tail-handled mice do not return to control levels whereas in both cases the tunnel handled mice do - I know that this is not reflected in the statistics (in terms of the 3 way interaction handling * contrast * phase) but it does perhaps suggest that there is more similarity in response to the contrast conditions than the authors suggest.

One option for revision could be for the authors to be more cautious in their claims and to more fully discuss the points I raise above - another would be to repeat the experiment but this may not be possible for all the obvious reasons -

Another issue is the question of whether the authors are distinguishing between mood and emotional states as they claim

In brief they use the handling treatments to establish a longer-term mood state which they validate with various tests/ measures

They then apply the contrast tests as measures of acute emotional responses ('disappointment'/ 'elation') - however they measure the contrast effect over a number of days (post-shift) and indeed with the negative contrast show longer-term effects of handling on disappointment (with a difference emerging between tail-handled vs tunnel handled emerging some days after the shift) - so my question is given that the contrast effect (as affected by handling) only emerges after some days post-shift can this be regarded as an acute emotional response?

Other points/ suggestions

Line 24: 'Knowing how animals experience emotions is challenging' - I would suggest that it is unlikely we will ever know how animals experience emotions

Lines 43-50: I would have thought that the prime reasons for investigating animal emotions was for the animals themselves and not for comparison to ourselves -

Line 53-54: change positive states to psychological states

Line 55: the quoted references do not provide direct support for this statement

Lines 51-65: I suggest referring to Fraser, D., & Duncan, I. J. (1998). 'Pleasures', 'pains' and animal welfare: toward a natural history of affect. *Animal welfare*, 7(4), 383-396.

Line 80: Refer to Burman's work on contrast effects?

Lines 136-137: As written the handling does not sound entirely standardised with mice potentially having different handling experiences depending on the requirements of their husbandry ('From that point, mice were only handled by their designated method, including for routine husbandry and transferring mice for behavioural testing')

Line 206-208: Explain the purpose of the setting up of testing cohorts

Line 266-269: As indicated above I see this as an 'overstatement' or alternatively not dealing sufficiently with the sucrose consumption issue

Fig 1D: The legend for sucrose consumption should be changed to be more understandable
Lick cluster as a measure looks more sensitive at lower sucrose concentration

Lines 306-314: I found that this paragraph was too dismissive of the lack of effect of handling on sucrose consumption

Referee: 2

Comments to the Author(s)

This paper describes a well-designed study of laboratory mice to investigate the relationship between an induced long-term negative affective state or 'mood' and shorter-term positive and negative affective states ('emotions') inferred from responses made in successive negative contrast (SNC) and successive positive contrast (SPC) tests. The interface between 'moods' and 'emotions' is a theoretically important area of animal emotion research and may have practical implications, including in an animal welfare context as the authors argue. The data are collected and analysed appropriately, the paper is clearly written and the results are interesting - the data provide convincing evidence of induction of the negative mood state, plus the suggestion that such a state may have a moderating effect on SNC and SPC responses. My main comments are on interpretation of results of the SNC and SPC tests and the potential for lack of differences during pre-shift baseline training and drift in the control groups during post-shift testing to have contributed to the findings (see below). I also have comments on clarifying concepts and terminology (always a potential source of confusion in studies of animal emotion), and some methodological details.

Main comments

Lines 213-216: A perennial challenge of SNC and SPC tests is to ensure that, at the time of the reward shift, groups trained on one reward are behaving in a clearly different way to those trained on the other (and hence that the rewards are valued differently), and also that control groups are showing a consistent response across sessions so that post-shift changes can be attributed to changes in treatment group behaviour rather than continuing drift in control groups. Did you check whether control group behaviour 'plateaued' during the pre-shift phase and was different to that of treatment groups? If not, what was the rationale for selecting a 10-day pre-shift period?

Lines 317-322: The analysis of SNC and SPC data don't appear to include a check on whether the groups trained on different sucrose concentrations within each comparison pair (Fig. 2A-D) show different lick-cluster sizes at the pre-shift (baseline) stage (see above). Although the results of the presented analysis of the pre-shift 'anhedonia test' (Fig. 1D) demonstrate a weak concentration effect, this looks to be primarily driven by the lower lick-cluster size in tail-handled mice exposed to 4% sucrose (though there is no significant interaction effect). Because Figs 2B-D all indicate very close pre-shift baselines, this point and its potential effects on interpretation must be considered in the Discussion.

Line 322-335: Post-hoc analyses of 3-way interactions can be done in several ways and one alternative here, related to the point made above about the possibility of 'control group response drift' following reward shift, would be to investigate whether control groups demonstrated different cross-time shifts in the different groups. For example, it looks from the graphs as if there was some upwards drift in lick cluster size in the tail-handled control group that was not so evident in the tunnel-handled control group, and this could contribute to the observed results. It would strengthen findings to rule this out, and the possibility that it influences findings should be considered in the Discussion.

Lines 344-349: The graphs indicate that tail-handled mice showed a more prolonged SPC than tunnel-handled mice (i.e. at post-shift phase 2; this might again be partly due to drift in control

group behaviour, e.g. downwards in tail-handled controls), so it is perhaps surprising to find no statistical effect. Given that there is also no 3-way interaction, it is difficult to justify further investigation of the data but some consideration of this possibility, including the influence of control group drift, would be useful to see in the Discussion.

Additional comments

Lines 33,35,401,403,413,457 and throughout: The word 'experience' as in 'experience discrete positive emotions' needs qualification because it suggests a conscious experience of the state and it is difficult / impossible to measure such conscious experiences in non-humans. Clarification of exactly how this term is being used in the paper will help avoid readers being confused about what is being proposed or inferred by the authors. The same comment applies for terms such as 'emotion', 'mood', 'depression', 'anxiety', 'disappointment' and 'elation' - a general statement about terminology is probably the most efficient way of clarifying usage (e.g. that these states are inferred from particular indicators, but that currently we cannot know for certain whether and how they are consciously experienced).

Lines 61-69: Please provide a brief explanation of why (theoretically and/or empirically) it is interesting to know about the interplay between 'mood' and 'emotion'. Specific hypotheses would also be useful if appropriate.

Lines 77-79: I think it would be more accurate to say here that the animal's discrete emotional responses can be inferred from measured responses to changes in reward value.

Lines 86-89: The 'discrete emotional response' doesn't indicate 'disappointment' or 'elation', it is the behavioural response that is used to infer these states which are themselves the discrete emotional responses. Again, this could be rephrased more clearly to say that particular behavioural responses to a gain or drop in reward value (the responses should be briefly described) can be interpreted as indicating a 'disappointment-like' or 'elation-like' state.

Line 90: Add a reference to a mouse SNC or SPC task.

Line 133: Please refer to the timeline of the study shown in Fig. 1A before starting to describe the tests etc. I was hoping for such a figure throughout the methods description as I tried to work out how everything fitted together!

Line 183: I assume that the 'sucrose drinking tests' refers to the 'contrast experiments'? If so, please keep terminology consistent.

Line 192: What days were the contrast experiments carried out on?

Line 221: The text says that there were 8 post-shift days, but Fig. 1A indicates only 4? In fact, it's not clear exactly when the 18 days of the contrast experiments occur on Fig. 1A. Please clarify.

Lines 222-223: Briefly explain why you divided these data into two bins, and what was the outcome measure in each bin (e.g. the mean of the responses shown in each 4-day block?).

Table 2: Please clarify what 'Anhedonia tests' refers to as there is no specific mention of this in the text or Fig. 1A. Is this the first (pre-shift) 10 days of the contrast experiments? If so, terminology should be used consistently to avoid confusion. Also, for the Open Field and Adrenal Weight measures, mice will have experienced not just the handling treatment but also the contrast treatment but this is not included as a factor in the statistical models - please provide justification of this.

Fig. 1B: It is interesting that the pre-handling time spent interacting with the hand is very low for tail-handled mice on day 1 even before the treatment has started. If this is correct, it needs to be considered in the Discussion (e.g. was there some (unintended/random) difference between

groups prior to the start of the study). If there is a good reason for this (e.g. handling was actually imposed before the start of the Affective State Manipulation period; mice are 'naturally' avoidant of human hands), this needs to be made clear.

Lines 276-281: Please add a reference to the Supplementary Materials to indicate that other results from the open field and physiological measures can be found there.

Lines 279-280: Clarify that the adrenal weight measure is a percentage of body weight, and that no significant difference in body weight was detected (Supplementary Materials).

Line 386: Might one expect different effects of anxiety and depression-like mood states on short-term emotions and, if so, could this also explain some results?

Author's Response to Decision Letter for (RSPB-2020-0030.R0)

See Appendix A.

RSPB-2020-1636.R0

Review form: Reviewer 2

Recommendation

Accept with minor revision (please list in comments)

Scientific importance: Is the manuscript an original and important contribution to its field?

Good

General interest: Is the paper of sufficient general interest?

Good

Quality of the paper: Is the overall quality of the paper suitable?

Good

Is the length of the paper justified?

Yes

Should the paper be seen by a specialist statistical reviewer?

No

Do you have any concerns about statistical analyses in this paper? If so, please specify them explicitly in your report.

No

It is a condition of publication that authors make their supporting data, code and materials available - either as supplementary material or hosted in an external repository. Please rate, if applicable, the supporting data on the following criteria.

Is it accessible?

Yes

Is it clear?

Yes

Is it adequate?

Yes

Do you have any ethical concerns with this paper?

No

Comments to the Author

I thank the authors for addressing my comments clearly and thoroughly, and am happy for the paper to be published. I have a couple of final minor comments:

Figure 1 legend: This is a very useful additional figure, but the colour of the line bars in the legend is difficult to see.

Figure 2 legend: I suggest that in the legend to panel A it is made clear that the interaction test differs between the treatment groups (i.e. tail-handled mice are exposed to a gloved human hand whilst tunnel-handled mice are exposed to a gloved hand holding a tunnel). Also, the legend for panel E indicates that lick cluster size is shown in the right panel, but the graph shows sucrose consumption.

Lines 911-914 (of track-changed version): "These findings were not exacerbated by drift among the control animals, since there was no significant difference in lick cluster sizes in the tail and tunnel control mice across the two post-shift phases". I think this analysis refers to the effect of trial on lick cluster size in control groups during post-shift phases 1 and 2. If so, best to make clear here that the independent variable was indeed trial.

Decision letter (RSPB-2020-1636.R0)

27-Jul-2020

Dear Dr Clarkson

I am pleased to inform you that your manuscript RSPB-2020-1636 entitled "Negative mood affects the expression of negative but not positive emotions in mice" has been accepted for publication in Proceedings B.

The referee(s) have recommended publication, but also suggest some minor revisions to your manuscript. Therefore, I invite you to respond to the referee(s)' comments and revise your manuscript. Because the schedule for publication is very tight, it is a condition of publication that you submit the revised version of your manuscript within 7 days. If you do not think you will be able to meet this date please let us know.

[http://datadryad.org/submit?journalID=RSPB&manu=\(Document not available\)](http://datadryad.org/submit?journalID=RSPB&manu=(Document not available)) which will take you to your unique entry in the Dryad repository. If you have already submitted your data to dryad you can make any necessary revisions to your dataset by following the above link.

Please see <https://royalsociety.org/journals/ethics-policies/data-sharing-mining/> for more details.

Sincerely,
Dr Sasha Dall
mailto: proceedingsb@royalsociety.org

Associate Editor

Comments to Author:

The reviewers' comments have generally been well addressed, and a few additional minor suggestions remain to improve the manuscript.

Reviewer(s)' Comments to Author:

Referee: 2

Comments to the Author(s).

I thank the authors for addressing my comments clearly and thoroughly, and am happy for the paper to be published. I have a couple of final minor comments:

Figure 1 legend: This is a very useful additional figure, but the colour of the line bars in the legend is difficult to see.

Figure 2 legend: I suggest that in the legend to panel A it is made clear that the interaction test differs between the treatment groups (i.e. tail-handled mice are exposed to a gloved human hand whilst tunnel-handled mice are exposed to a gloved hand holding a tunnel). Also, the legend for panel E indicates that lick cluster size is shown in the right panel, but the graph shows sucrose consumption.

Lines 911-914 (of track-changed version): "These findings were not exacerbated by drift among the control animals, since there was no significant difference in lick cluster sizes in the tail and tunnel control mice across the two post-shift phases". I think this analysis refers to the effect of trial on lick cluster size in control groups during post-shift phases 1 and 2. If so, best to make clear here that the independent variable was indeed trial.

Author's Response to Decision Letter for (RSPB-2020-1636.R0)

See Appendix B.

Decision letter (RSPB-2020-1636.R1)

29-Jul-2020

Dear Dr Clarkson

I am pleased to inform you that your manuscript entitled "Negative mood affects the expression of negative but not positive emotions in mice" has been accepted for publication in Proceedings B.

Open Access

Paper charges

Sincerely,

Appendix A

Response to Editorial and Reviewers' comments

Associate Editor

Board Member

Both reviewers found the manuscript well presented and of good general interest. They have identified several key areas that require attention, in particular: further clarification of concepts and terminology; more cautious and more detailed consideration of the results in relation to previous studies; and potential additional statistical analysis to consolidate interpretation of the study findings.

We're delighted that both reviewers enjoyed reading our manuscript, and saw its potential for advancing what we know about animal emotions. We'd like to thank both reviewers for their really considered and thoughtful comments on our paper: they have been invaluable in helping us reflect further on our findings and presentation. We have addressed all their main points, and paid particular attention to those relating to our terminology and interpretation of our results, and have added new analyses, which we hope the reviewers agree strengthen our results. Furthermore, given the inclusion of additional analyses we edited the full text in order to meet journal formatting requirements, which are also reflected in the tracked changes.

NB. New cited line numbers included in the response to reviewers below, refer to line numbers for the clean version of the resubmitted manuscript and not the tracked version.

Reviewer(s)' Comments to Author

Referee 1

This paper addresses a very interesting question - the relationship between longer-term mood and expression of discrete negative/ positive emotions - the paper could be of considerable general interest - the paper's approach is novel and potentially exciting

We'd like to thank the reviewer for these positive comments highlighting the strength of our paper, as well as their challenging and thoughtful comments on our manuscript which we fully address below. We have found them immensely helpful in revising and improving our manuscript, and we now include additional analyses and explanation, as requested. We hope that the reviewer is now satisfied with our treatment and interpretation of our data.

However I am concerned that the authors are somewhat overstating their findings at least with respect to the claim that they have unequivocally demonstrated that their handling treatments have given rise to expected changes in anhedonic responses - here their results vary in one important respect from their (& others) previous work in that the mice did not reduce their consumption of sucrose as expected.

In response to this the authors turn to another measure (lick cluster size) as their measure of anhedonia which allows them to proceed to claim that the tail-handling treatment induced anhedonia - however as the authors state themselves (lines 271-273) previous work on anhedonia using the sucrose model finds both a reduction in level of sucrose drunk as well as changes in lick cluster size.

Given this dissociation of these 2 measures of anhedonia I think we have to view the remaining results on contrast effects with some caution as we cannot be certain of the baseline condition (i.e. whether the tail-handled mice were indeed anhedonic) - the authors do acknowledge some of these issues lines 306-314; here in effect they 'bin' sucrose consumption as a measure of contrast effects because it does not fit their expectations ('consumption did not accurately reflect the difference in reward value'); in doing so they are choosing to not fully consider both their own and others previous findings where tail-handling led to a reduction in sucrose consumption.

We fully understand the reviewer's concerns here, and that our interpretation appears selective from the data we present (which was not our intent). We had our reasons for thinking that lick cluster size may be a more reliable measure of hedonic evaluation, as we explain below, but we perhaps did not make these sufficiently clear in the paper. These comments also prompted us to go back to our original data and conduct an additional analysis that shows that tail handled mice drank less sucrose in both post-shift phases compared to tunnel handled mice. We think that this new finding along with our changes to the manuscript strengthen the case for mice being anhedonic, and we hope that our more detailed interpretation and discussion of the results allays the reviewer's concern.

First of all, we want to explain our original interpretation: that differences in lick cluster sizes show that mice are anhedonic, even in the absence of supporting sucrose consumption data. The reviewer is absolutely right in saying that sucrose consumption is a well-established measure of anhedonia and depression in mice (e.g. see review Willner P. *Neurobiology of Stress* 2017;(6):78-93). However, consumption can also be affected by motivation, meaning that it is sensitive to caloric intake and satiety, particularly at higher sucrose concentrations like the ones we used (Collier G, Bolles R. *J Comp Physiol Psychol.* 1968;65(3):379–83); Pfaffmann C. *Am J Clin Nutr.* 1957;5:142–7; Richter CP, Campbell KH. *J Nutr.* 1940 Jul 1;20(1):31–46). This is already evident in the sucrose consumption data we present for the pre-shift phase, where mice in both groups consume less 32% sucrose than 4% sucrose solution, despite 32% being sweeter and predicted to be drunk more. This is easily explained from other studies (see above references) mice drink more calories more quickly with 32% compared to 4% sucrose, and as a consequence, become satiated more quickly and drink less overall.

In contrast, lick cluster size, by being a direct measure of how mice respond to a sweet taste through the way they drink, is thought to be a more sensitive measure of hedonic valuation (Dwyer D; *Q J Exp Psychol,* 2012;65(3):371-394). Therefore, because our consumption data could be (and was) affected by motivational factors, and that lick cluster sizes were smaller in tail handled mice, we thought it safe to conclude that tail handled mice were more anhedonic at the end of the pre-shift phase. However, we fully take on board the reviewer's point that we need to reflect on this more, and we now provide further discussion of our two measures and our interpretation of them to reflect the data we have.

Second, the reviewer's comments made us reflect on whether we had additional sucrose consumption data from another part of the experiment that could further support the view that our tail handled mice were anhedonic. So far, we had only considered the data at the end of the pre-shift phase: after this point, mice received their experimental treatments, and we could not include mice that now experienced a different concentration of sucrose. However, all our control mice (N=32) continued to receive the same concentration of sucrose in the two post-shift phases, so we looked for evidence of differences in hedonic experience between tail and tunnel handled mice in these two later phases (N=8 in each). Although this subset of mice also didn't show evidence for differences in sucrose consumption in the pre-shift phase, we did find that in both post-shift phases, tail handled mice drank less sucrose and were more anhedonic than tunnel handled mice (left panel: post-shift 1 phase; right panel: post-shift 2 phase).

We now include this figure (Figure 2F) and associated analysis in our results section (lines: 274-288) and supplementary materials (section 1.2.7). We hope the reviewer agrees that this adds further evidence that our tail handled mice showed anhedonia, a core symptom of depression, during our experimental testing. We cannot fully explain why we do not see this difference in sucrose consumption in the pre-shift phase, but reflect on the evidence that our mice were anhedonic in the Discussion (lines 426-446).

One further point on this - eye-balling the response curves to both the negative and positive contrasts the shape of the response is similar for both contrasts and both handling treatments - in both cases visually the tail-handled mice do not return to control levels whereas in both cases the tunnel handled mice do - I know that this is not reflected in the statistics (in terms of the 3 way interaction handling * contrast * phase) but it does perhaps suggest that there is more similarity in response to the contrast conditions than the authors suggest. One option for revision could be for the authors to be more cautious in their claims and to more fully discuss the points I raise above - another would be to repeat the experiment but this may not be possible for all the obvious reasons

We agree that eyeballing what is now Figure 3 could lead a reader to think this, but as the Reviewer themselves point out, our statistical analyses do not support this interpretation. In addition, it is important to note that the positively shifted tunnel mice (blue line; Figure 3D) do not change their behaviour across the two post-shift phases. This reviewer's interpretation that the results appear more similar than we acknowledge is only possible because of a possible drift in the control SPC group in the tunnel handled mice in the post-shift phase (grey line, Figure 3D). However, we find no evidence of drift (see response to Reviewer 2 below, comment 344-349, and now reported in the manuscript lines 329-334) and have no sound basis for making this interpretation.

Another issue is the question of whether the authors are distinguishing between mood and emotional states as they claim. In brief they use the handling treatments to establish a longer-term mood state which they validate with various tests/ measures. They then apply the contrast tests as measures of acute emotional responses ('disappointment'/ 'elation') - however they measure the contrast effect over a number of days (post-shift) and indeed with the negative contrast show longer-term effects of handling on disappointment (with a difference emerging between tail-handled vs tunnel handled emerging some days after the shift) - so my question is given that the contrast effect (as affected by handling) only emerges after some days post-shift can this be regarded as an acute emotional response?

This is an excellent question about how our contrast effects, measured over a number of days, can capture an acute emotional response. To help answer this, we draw on the extensive experimental psychology literature on contrast effects, which use similar methodology to measure short-term responses towards increasing or decreasing reward value (referred to as 'elation', 'frustration' and 'disappointment') over a number of trials/days (e.g. Benfield et al. 1974, J. Comp.

Phys. Psych. 86:848; Chen et al. 1981, J. Comp. Phys. Psych. 95:146; Flaherty et al. 1983, Anim. Learn. Behav. 11:407; Pellegrini & Mustaca 2000, Learn. Motiv. 31:200; Ellis et al. 2020, J. Appl. Anim. Welf. Sci. 23:54). The key thing is that our study, like those in this literature, measures acute emotional responses in a short trial, but does so repeatedly. Therefore, each trial gives a measure of an acute response, but the repeated trials gives a measure of the longevity of that acute response, i.e. how long it is sustained. As the reviewer quite rightly points out, our difference in the handling group emerges in the SNC after a number of trials, showing that tail handled mice remain disappointed for longer than tunnel handled mice. We perhaps did not make this clear enough, and have hopefully now referred to this better in our manuscript.

Other points/ suggestions

Line 24: 'Knowing how animals experience emotions is challenging' - I would suggest that it is unlikely we will ever know how animals experience emotions

We fully agree. We have now rephrased this to 'whether and to what extent animals experience emotions is challenging'(line 24).

Lines 43-50: I would have thought that the prime reasons for investigating animal emotions was for the animals themselves and not for comparison to ourselves –

As researchers with an interest in animal welfare, we agree with the reviewer (although researchers in other fields may not!). We have added the phrase, 'and as importantly' to emphasise the importance for animal welfare and direct benefits to animals (line 48).

Line 53-54: change positive states to psychological states

We weren't quite sure what was meant by the term 'psychological states', particularly given that psychological states need not be positive and were keen to minimise the number of different terms we use. Therefore, we changed 'positive states' to 'positive emotional states' (line 59), and hope that this addresses the reviewer's concern.

Line 55: the quoted references do not provide direct support for this statement

We have now added additional references providing more direct support (lines 61-63).

Lines 51-65: I suggest referring to Fraser, D., & Duncan, I. J. (1998). 'Pleasures','pains' and animal welfare: toward a natural history of affect. Animal welfare, 7(4), 383-396.

Now cited appropriately in the text (line 67).

Line 80: Refer to Burman's work on contrast effects?

Now cited with the appropriate references (line 89 onwards).

Lines 136-137: As written the handling does not sound entirely standardised with mice potentially having different handling experiences depending on the requirements of their husbandry ('From that point, mice were only handled by their designated method, including for routine husbandry and transferring mice for behavioural testing')

To clarify, the handling was entirely standardised: all mice had the same number of handling sessions and all mice were cleaned out once per week. We have now clarified this in the text (lines 129-139).

Line 206-208: Explain the purpose of the setting up of testing cohorts

This was because we only had eight drinking cages and as such the animals had to be tested in four batches. Therefore we needed to set up testing cohorts to ensure that the different

treatment groups were fully balanced with regards to equal numbers of tail and tunnel handled mice in each testing batch. We have clarified this short paragraph (lines 202-206).

Line 266-269: As indicated above I see this as an 'overstatement' or alternatively not dealing sufficiently with the sucrose consumption issue

This comment referred to our presentation of the evidence that our mice were anhedonic. In light of further analyses, we have re-written this section (lines 274-288), and added consumption data that strengthen the evidence that our tail handled mice were anhedonic (see response in full above). We hope that reviewer agrees that we have responded in full to this concern.

Fig 1D: The legend for sucrose consumption should be changed to be more understandable Lick cluster as a measure looks more sensitive at lower sucrose concentration

We think that the reviewer is referring to the fact that we omitted to refer to the right panel in the legend, which we have now corrected. Whilst lick cluster size may look more sensitive at lower sucrose concentration, we do not find an interaction between handling method and sucrose concentration (GLM reported in full in the supplementary materials section 1.2.6), and we did not make any change in response to this comment.

Lines 306-314: I found that this paragraph was too dismissive of the lack of effect of handling on sucrose consumption

Re-reading the manuscript we understand the reviewer's concern here: it was not our intention to be dismissive. What we wanted to point out was that we could not look at consumption in the contrast experiments because the measure was not reflecting hedonic value: mice were drinking more 4% than 32% sucrose solution. This is explained when we consider the satiating properties of the two solutions, with 32% sucrose leading to a faster calorific intake and the need for less consumption compared to 4% sucrose. We have now hopefully explained this better, and combined it with results from a couple of other tests requested by the second reviewer under a new heading in the main text: 'Contrast Experiments' and in the supplementary materials sections 1.2.5-1.3.2). This hopefully better explains our approach to data handling and analysis

Referee: 2

This paper describes a well-designed study of laboratory mice to investigate the relationship between an induced long-term negative affective state or 'mood' and shorter-term positive and negative affective states ('emotions') inferred from responses made in successive negative contrast (SNC) and successive positive contrast (SPC) tests. The interface between 'moods' and 'emotions' is a theoretically important area of animal emotion research and may have practical implications, including in an animal welfare context as the authors argue. The data are collected and analysed appropriately, the paper is clearly written and the results are interesting - the data provide convincing evidence of induction of the negative mood state, plus the suggestion that such a state may have a moderating effect on SNC and SPC responses. My main comments are on interpretation of results of the SNC and SPC tests and the potential for lack of differences during pre-shift baseline training and drift in the control groups during post-shift testing to have contributed to the findings (see below). I also have comments on clarifying concepts and terminology (always a potential source of confusion in studies of animal emotion), and some methodological details.

We appreciate the reviewer's positive comments on the design, analysis and writing up of our study. We understand that we could have been clearer in our terminology and interpretation, and address the reviewer's valuable comments below.

Main comments

Lines 213-216: A perennial challenge of SNC and SPC tests is to ensure that, at the time of the reward shift, groups trained on one reward are behaving in a clearly different way to those trained on the other (and hence that the rewards are valued differently), and also that control groups are showing a consistent response across sessions so that post-shift changes can be attributed to changes in treatment group behaviour rather than continuing drift in control groups. Did you check whether control group behaviour ‘plateaued’ during the pre-shift phase and was different to that of treatment groups? If not, what was the rationale for selecting a 10-day pre-shift period?

We would like to thank the reviewer for highlighting these challenges associated with running meaningful contrast paradigms. We have now added a new section in our Results (lines 311-334)(supported by supplementary material section 1.3) that specifically addresses these challenges.

The 10-day pre-shift phase was to ensure that our mice acclimatised to the drinking chamber, and we can confirm that all of our control groups were showing a consistent response across sessions and plateauing in the final four trials of the pre-shift phase (full analyses now included in the supplementary material section 1.3). None of our control groups showed any significant drift during the post-shift phase, and we have also now added these analyses to the paper (lines 353-356).

Our lick cluster size data for all mice (presented in Figure 2E, Table S4; Figure S3) shows that they value the higher (32% sucrose) reward more than the lower (4% sucrose) reward at the end of the pre-shift period. The query regarding whether control groups’ behaviour differed from that of their treatment groups at the end of the pre-shift is addressed in our answer to the next comment which expands on this question.

Lines 317-322: The analysis of SNC and SPC data don’t appear to include a check on whether the groups trained on different sucrose concentrations within each comparison pair (Fig. 2A-D) show different lick-cluster sizes at the pre-shift (baseline) stage (see above). Although the results of the presented analysis of the pre-shift ‘anhedonia test’ (Fig. 1D) demonstrate a weak concentration effect, this looks to be primarily driven by the lower lick-cluster size in tail-handled mice exposed to 4% sucrose (though there is no significant interaction effect). Because Figs 2B-D all indicate very close pre-shift baselines, this point and its potential effects on interpretation must be considered in the Discussion.

Since we had established that mice tail and tunnel handled mice differed in their lick cluster sizes at the pre-shift phase, we had not included these analyses within each comparison pair (Now Fig. 3A-D). It is important to note that because the experiment was conducted in two batches, this could only be done once all the data had been collected in order to provide sufficient sample sizes for analysis (N=4 in each group in each batch). Therefore, establishing this as part of our experimental design was not possible, but we agree with the reviewer that conducting this analysis is relevant.

We have now described these analyses in our results (lines 322-328) and in full in our supplementary materials (section 1.3.1), and as the reviewer observes, control and treatment groups did not always differ in their lick cluster sizes (Figs 3B-D). Of course, our experiment differs from standard SNC/SPC protocols in that we manipulate and create differences in the hedonic responses of mice prior to the contrast experiments, and we know from earlier studies that as a consequence, lick cluster sizes of tunnel handled mice appear more similar to low and high value rewards (see also Clarkson et al. 2018). Therefore, we didn’t necessarily expect these differences that the reviewer expects to see.

However, we don’t think having close pre-shift baselines negates the contrast effects that we see, especially as our contrast comparisons focus on differences in lick cluster sizes across the two post-shift phases. Showing this difference in the pre-shift is important for interpreting negative results in the post-shift phase: if there is no contrast effect, could it simply be that the animals didn’t detect a change in reward value or value the two concentrations differently. This is not a concern in our experiment since all groups reliably showed significant contrast effects.

In summary, we agree that readers may want to see these comparisons, and have now conducted and included these analyses for completeness. However, as we discuss above, we don't think that they negate our findings. We also agree with the reviewer that we should discuss this, and have included an additional paragraph about our methodology in the Discussion (paragraph starting line 426-446).

Line 322-335: Post-hoc analyses of 3-way interactions can be done in several ways and one alternative here, related to the point made above about the possibility of 'control group response drift' following reward shift, would be to investigate whether control groups demonstrated different cross-time shifts in the different groups. For example, it looks from the graphs as if there was some upwards drift in lick cluster size in the tail-handled control group that was not so evident in the tunnel-handled control group, and this could contribute to the observed results. It would strengthen findings to rule this out, and the possibility that it influences findings should be considered in the Discussion.

We agree that such post-hoc tests can be performed a number of ways, and in our manuscript, we focussed on those most relevant to our research questions. However, we agree that it is important to rule out effects of drift in these two control groups to strengthen our findings. Paired t-tests between Postshift 1 and Postshift 2 phases for each control group (Tail SNC Con and Tunnel SNC Con) were both non-significant, and we now report this in our Results (line 322-328) and Discussion (Lines 426-446).

Lines 344-349: The graphs indicate that tail-handled mice showed a more prolonged SPC than tunnel-handled mice (i.e. at post-shift phase 2; this might again be partly due to drift in control group behaviour, e.g. downwards in tail-handled controls), so it is perhaps surprising to find no statistical effect. Given that there is also no 3-way interaction, it is difficult to justify further investigation of the data but some consideration of this possibility, including the influence of control group drift, would be useful to see in the Discussion.

We agree that without a three-way interaction, we can not justify further detailed investigation of the data. We have now conducted the analyses around the possibility of drift (line 329-334) where there is a possibility that it could influence any interpretation of our results (see previous comment).

Additional comments

Lines 33,35,401,403,413,457 and throughout: The word 'experience' as in 'experience discrete positive emotions' needs qualification because it suggests a conscious experience of the state and it is difficult / impossible to measure such conscious experiences in non-humans. Clarification of exactly how this term is being used in the paper will help avoid readers being confused about what is being proposed or inferred by the authors. The same comment applies for terms such as 'emotion', 'mood', 'depression', 'anxiety', 'disappointment' and 'elation' - a general statement about terminology is probably the most efficient way of clarifying usage (e.g. that these states are inferred from particular indicators, but that currently we cannot know for certain whether and how they are consciously experienced).

We thank the reviewer for highlighting this; we did not want to infer conscious experience in the language we used, and have now largely removed this term throughout the manuscript. We continue to use it when talking more broadly about the uncertainties of how emotions might be experienced by animals. We think this reduces ambiguity and confusion for readers.

Lines 61-69: Please provide a brief explanation of why (theoretically and/or empirically) it is interesting to know about the interplay between 'mood' and 'emotion'. Specific hypotheses would also be useful if appropriate.

We did discuss why we considered it important to focus on the interplay between emotion and mood in the first half of the paragraph that the reviewer refers to, however, we appreciate that we weren't perhaps explicit enough. Essentially, it's important to know how short-term emotions contribute to long-term mood changes, as well how long term moods affect short-term emotional responses towards lifetime events. Specific hypotheses are difficult because data are rather limited, with models of animal emotion based around those developed for humans. Our study emerge from what is known from studies of MDD, that people in a depressed mood report stronger negative emotions but more but blunted positive emotional responses. We have now re-written this paragraph (line 59-76) to be more explicit, and we hope that this is now clearer. This also led to changes in the following paragraphs that we hope explain the value in our approach.

Lines 77-79: I think it would be more accurate to say here that the animal's discrete emotional responses can be inferred from measured responses to changes in reward value.

Agree, this has now been changed in the text.

Lines 86-89: The 'discrete emotional response' doesn't indicate 'disappointment' or 'elation', it is the behavioural response that is used to infer these states which are themselves the discrete emotional responses. Again, this could be rephrased more clearly to say that particular behavioural responses to a gain or drop in reward value (the responses should be briefly described) can be interpreted as indicating a 'disappointment-like' or 'elation-like' state.

Agree, this is now rephrased to be clearer, including reference to typical behavioural responses seen with the literature to clarify (lines 98-101).

Line 90: Add a reference to a mouse SNC or SPC task.

Now added.

Line 133: Please refer to the timeline of the study shown in Fig. 1A before starting to describe the tests etc. I was hoping for such a figure throughout the methods description as I tried to work out how everything fitted together!

Thanks for highlighting this, we have now added an appropriate timeline figure (Figure 1) to highlight the timing of each of these tests in the method section.

Line 183: I assume that the 'sucrose drinking tests' refers to the 'contrast experiments'? If so, please keep terminology consistent.

Yes. We have now change to 'contrast experiments' throughout.

Line 192: What days were the contrast experiments carried out on?

The contrast experiments were carried out on days 15-24 for the 10 pre-shift trials and days 25-32 for the 8 post-shift trials. We have now clarified this (Figure 1).

Line 221: The text says that there were 8 post-shift days, but Fig. 1A indicates only 4? In fact, it's not clear exactly when the 18 days of the contrast experiments occur on Fig. 1A. Please clarify.

We apologise for the confusion, and have now clarified this in the study timeline figure (Figure 1), as suggested by the reviewer (see above).

Lines 222-223: Briefly explain why you divided these data into two bins, and what was the outcome measure in each bin (e.g. the mean of the responses shown in each 4-day block?).

We pooled the data into bins of 4 days because lick cluster size can vary across trials. We took the mean lick cluster size across the four trials as the outcome measure in each bin. We have now clarified this in the text (lines 229-233).

Table 2: Please clarify what ‘Anhedonia tests’ refers to as there is no specific mention of this in the text or Fig. 1A. Is this the first (pre-shift) 10 days of the contrast experiments? If so, terminology should be used consistently to avoid confusion.

We apologise: the use of the term ‘anhedonia tests’ was confusing and not consistent with the rest of the text (we were referring to the drinking data from last four preshift trials). This term has been removed to avoid confusion, and we have clarified Table 2 (which is now Table S1 in the supplementary materials).

Also, for the Open Field and Adrenal Weight measures, mice will have experienced not just the handling treatment but also the contrast treatment but this is not included as a factor in the statistical models – please provide justification of this.

First of all, we should highlight that the adrenal glands were only taken from control mice which did not experience any contrast treatment. Because no-one has measured chronic physiological effects of tail handling in mice before, we chose to use control mice to avoid any possible effects from our different contrast treatments. We have highlighted this more in the text to avoid any confusion (lines 239-241).

However, the reviewer is correct in saying that we used all our mice in our OF test following the contrast treatments. This test was conducted to check that our tail and tunnel handled mice still differed in their anxiety levels, and make the data comparable to the EPM test by using all mice, not a subset. Therefore, showing any effect of contrast treatment on anxiety was not expected or of interest, which is why we did not include it as a factor in our models. Indeed, it would be difficult to include contrast treatment as a factor, because in some mice, the effect of contrast treatment was positive and for others it was negative. Since we find a difference using all the mice, regardless of contrast treatment, we prefer to report this to simplify our results.

Fig. 1B: It is interesting that the pre-handling time spent interacting with the hand is very low for for tail-handled mice on day 1 even before the treatment has started. If this is correct, it needs to be considered in the Discussion (e.g. was there some (unintended/random) difference between groups prior to the start of the study). If there is a good reason for this (e.g. handling was actually imposed before the start of the Affective State Manipulation period; mice are 'naturally' avoidant of human hands), this needs to be made clear.

Now Fig 2B: We agree that this is an interesting effect, but it has been seen in a number of previous handling studies using the same voluntary interaction tests (Hurst et al 2010; Hurst et al 2013; Clarkson et al 2018). Mice are obtained from a commercial breeder, where they will have been handled a small number of times previously. Animals are randomly allocated to treatment groups, and there can be no unintended/random differences between our groups (see also Hurst et al 2010; Hurst et al 2013; Clarkson et al 2018). The difference between handling groups, even in the first voluntary interaction test is most likely due to animals are more willing to approach a tunnel than a hand based on their previous experience. The data from these tests were collected to check that our handling manipulation produced the same behaviours as seen in previous studies, and not as a validated measure of anxiety or depression. We do not see any reason to discuss this further in the Discussion.

Lines 276-281: Please add a reference to the Supplementary Materials to indicate that other results from the open field and physiological measures can be found there.

Now added.

Lines 279-280: Clarify that the adrenal weight measure is a percentage of body weight, and that no significant difference in body weight was detected (Supplementary Materials).

Done. We’ve now stated that adrenal weights were relative to bodyweight, and directed readers to the Supplementary Materials (line 268; 271).

Line 386: Might one expect different effects of anxiety and depression-like mood states on short-term emotions and, if so, could this also explain some results?

This is an interesting question, and one we have thought hard about. It is certainly theoretically possible that long-term anxiety and depression could cause different effects on short-term emotions. We're sure that the reviewer will be very familiar with the leading models in this field (Mendl et al 2010; Nettle & Bateson 2012), which predict that sensitivity to positive events will differ between animals that are either anxious or depressed. However, our tail handled animals appear to be both more anxious and more depressed than our tunnel handled mice, making it impossible to disassociate the effects of the two in the interpretation of the results. Indeed, anxiety is commonly comorbid with depression in humans. Therefore, we can't see how this might help to explain some results, and have not made any changes to the manuscript.

Appendix B

Associate Editor

Comments to Author:

The reviewers' comments have generally been well addressed, and a few additional minor suggestions remain to improve the manuscript.

We would like to thank both reviewers and the editor for their helpful comments and time reviewing our paper. We have now addressed the remaining minor comments below prior to publication.

Reviewer(s)' Comments to Author:

Referee: 2

Comments to the Author(s).

I thank the authors for addressing my comments clearly and thoroughly, and am happy for the paper to be published. I have a couple of final minor comments:

We are delighted you found our revisions clear and thorough and would like to thank you for your helpful comments on our paper. We have now addressed your additional comments below.

Figure 1 legend: This is a very useful additional figure, but the colour of the line bars in the legend is difficult to see.

We have now added text instead of the colour line bars to signify each component of the figure.

Figure 2 legend: I suggest that in the legend to panel A it is made clear that the interaction test differs between the treatment groups (i.e. tail-handled mice are exposed to a gloved human hand whilst tunnel-handled mice are exposed to a gloved hand holding a tunnel). Also, the legend for panel E indicates that lick cluster size is shown in the right panel, but the graph shows sucrose consumption.

We have now clarified the differences between the voluntary interaction tests in the figure legend and amended panel E so that the left panel illustrates consumption and the right panel illustrates lick cluster size, thank you for pointing out this error.

Lines 911-914 (of track-changed version): "These findings were not exacerbated by drift among the control animals, since there was no significant difference in lick cluster sizes in the tail and tunnel control mice across the two post-shift phases". I think this analysis refers to the effect of trial on lick cluster size in control groups during post-shift phases 1 and 2. If so, best to make clear here that the independent variable was indeed trial.

The analysis we did here was looking at post-shift phase (1 and 2) which was made up of four independent trials each. Therefore, the independent variable was post-shift phase rather than trial. We have now clarified in the text to avoid confusion. "These findings were not exacerbated by drift among the control animals, since there was no significant

difference in lick cluster sizes across the two post-shift phases for tail and tunnel control mice" (line 367-370).